# Development and Regulation of the Extreme Biofilm Formation of *Deinococcus radiodurans* R1 under Extreme Environmental Conditions

**DOI:** 10.3390/ijms25010421

**Published:** 2023-12-28

**Authors:** Qiannan Guo, Yuhua Zhan, Wei Zhang, Jin Wang, Yongliang Yan, Wenxiu Wang, Min Lin

**Affiliations:** 1National Key Laboratory of Agricultural Microbiology, Biotechnology Research Institute, Chinese Academy of Agricultural Sciences, Beijing 100081, China; guoqiannan0709@163.com (Q.G.); zhanyuhua@caas.cn (Y.Z.); zhangwei01@caas.cn (W.Z.); wangjin@caas.cn (J.W.); yanyongliang@caas.cn (Y.Y.); wenbrwang@163.com (W.W.); 2Key Laboratory of Agricultural Microbiome (MARA), Biotechnology Research Institute, Chinese Academy of Agricultural Sciences, Beijing 100081, China

**Keywords:** extreme environment, *Deinococcus radiodurans*, biofilm formation, transcriptome profile, response regulator DrRRA

## Abstract

To grow in various harsh environments, extremophiles have developed extraordinary strategies such as biofilm formation, which is an extremely complex and progressive process. However, the genetic elements and exact mechanisms underlying extreme biofilm formation remain enigmatic. Here, we characterized the biofilm-forming ability of *Deinococcus radiodurans* in vitro under extreme environmental conditions and found that extremely high concentrations of NaCl or sorbitol could induce biofilm formation. Meantime, the survival ability of biofilm cells was superior to that of planktonic cells in different extreme conditions, such as hydrogen peroxide stress, sorbitol stress, and high UV radiation. Transcriptome profiles of *D. radiodurans* in four different biofilm development stages further revealed that only 13 matched genes, which are involved in environmental information processing, carbohydrate metabolism, or stress responses, share sequence homology with genes related to the biofilm formation of *Escherichia coli*, *Pseudomonas aeruginosa*, and *Staphylococcus aureus*. Overall, 64% of the differentially expressed genes are functionally unknown, indicating the specificity of the regulatory network of *D. radiodurans*. The mutation of the *drRRA* gene encoding a response regulator strongly impaired biofilm formation ability, implying that DrRRA is an essential component of the biofilm formation of *D. radiodurans*. Furthermore, transcripts from both the wild type and the *drRRA* mutant were compared, showing that the expression of *drBON1* (*Deinococcus radiodurans*
BON domain-containing protein 1) significantly decreased in the *drRRA* mutant during biofilm development. Further analysis revealed that the *drBON1* mutant lacked the ability to form biofilm and DrRRA, and as a facilitator of biofilm formation, could directly stimulate the transcription of the biofilm-related gene *drBON1*. Overall, our work highlights a molecular mechanism mediated by the response regulator DrRRA for controlling extreme biofilm formation and thus provides guidance for future studies to investigate novel mechanisms that are used by *D. radiodurans* to adapt to extreme environments.

## 1. Introduction

Biofilms are multicellular communities of microbes that are encased within a matrix of self-produced extracellular polymeric substances (EPSs), which are also known as “cities of microbes” [1,2]. In general, biofilm EPSs consist of exopolysaccharides, proteins, and extracellular DNA (eDNA) [3]. Microbial biofilm structures have been identified in various natural environments, such as deep-sea vents [4], hot springs [5], and extremely pathogenic or human-specific serovars [6]. Moreover, previous studies have shown that the ability to form biofilms can protect microorganisms from extreme environments [7,8,9,10]. Investigations have shown that extremophile *Deinococcus geothermalis* cells enclosed in biofilms are more tolerant than planktonic cells under simulated space conditions, Martian conditions, and space environments [11,12]. Biofilm formation is a multistage process that initially involves surface attachment (I) with subsequent microcolony formation (II), biofilm maturation (III), and concerted regulation combining both environmental signals and regulatory networks [2,13,14,15]. Numerous studies have established that regulatory circuits governing the transition from planktonic to biofilm environments are very complex and differ between microbial species, although common regulatory mechanisms such as the two-component system (TCS) exist [2,13,14,15,16]. For instance, the genetic elements and mechanisms underlying biofilm development of the Gram-positive opportunistic pathogen *Staphylococcus aureus,* which causes chronic biofilm-associated infections, in addition to the quorum-sensing system Agr, Sar family regulators, and the alternative sigma factor SigB [15,17], the TCSs (WalR/K [18], SaeR/S [19], SrrA/B [20], ArlR/S [21], LytR/S [22], GraR/S [23] and NsaR/S [24]) play key regulatory roles in *S. aureus* biofilm development. The TCS is a major strategy for connecting changes in input environmental signals to cellular physiological output and is composed of histidine kinase (HK) and response regulator (RR) [25]. Moreover, several TCSs, BfiSR, BfmSR, and MifSR, comprise a network to sequentially regulate during *Pseudomonas aeruginosa* biofilm formation [26]. To date, there have been few studies on the molecular mechanism of extreme biofilm formation. A recent study showed that stress response genes and EPS component saccharide biosynthesis genes are upregulated in a strong biofilm-overproducing mutant strain of the extremophile *Deinococcus metallilatus* MA1002, which is generated via UV mutagenesis [27].

*Deinococcus radiodurans* R1 is unparalleled among all known microorganisms regarding extreme resistance to ionizing radiation, UV light, toxic chemicals, and desiccation [28,29]. Understanding the adaptive capacities of these bacteria is crucial to revealing how they persist and remain active in such extreme conditions that are lethal to most organisms. It has previously been shown that the genome of the poly-extremophile *D. radiodurans* contains a set of significant stress response-related genes (namely, *drRRA*, *irrE*, *dtrA*, *dtrB*, *pprM*, and *carD*). The response regulator DrRRA plays a crucial role in stress resistance, and the *drRRA* mutant is sensitive to ionizing radiation, ultraviolet radiation, hydrogen peroxide, and desiccation [30]. Both IrrE and PprM are known as global regulators of radiation resistance and the oxidative stress response [31,32,33]. PprM regulates the catalase KatE1, the synthesis of deinoxanthin, and the concentration of metal ions in *D. radiodurans* [32,33,34]. A recombinant *D. radiodurans* strain has the capacity for heavy metal (Co, Ni) biosorption, and its uranyl ion removal efficiency of biofilm (75 ± 2%) is four times that of planktonic cells; therefore, *D. radiodurans* biofilm has great potential in the bioremediation of radioactive waste components [35,36]. However, no promising candidate regulators involved in the extreme biofilm formation of *D. radiodurans* have been identified thus far. In addition, most of the reported genes encoding biofilm-related molecules present in model biofilm-forming bacteria are absent from the *D. radiodurans* genome based on sequence alignment and homology search, strongly suggesting that the regulatory mechanism of biofilm formation in this bacterium is distinctive.

In this study, we explored the features and genetic elements of biofilm development in the extremophile *D. radiodurans* R1 and tested the stress resistance of biofilm cells. The results showed that abiotic stress (NaCl or sorbitol) induces biofilm formation quickly. In addition, cells in biofilms are more tolerant than planktonic cells under adverse environmental conditions, including H_2_O_2_ stress, UV radiation, and sorbitol stress. The differentially expressed genes based on the transcriptome analysis of cell samples are listed. Furthermore, the *drRRA* mutant and the *drBON1* mutant lost biofilm formation ability. Quantitative real-time PCR and electrophoretic mobility shift assay results showed that DrRRA regulates biofilm formation via transcriptional activation of *drBON1*.

## 2. Results

### 2.1. NaCl and Sorbitol Stress Induce Biofilm Formation of D. radiodurans

The capacities of organisms to respond to fluctuations in their osmotic environments are important physiological adaptations that determine their abilities to thrive in a variety of habitats. To explore whether biofilm formation is a kind of response strategy to extreme environmental conditions, the biofilm formation ability of *D. radiodurans* was examined under extremely high sodium chloride (NaCl) and sorbitol concentration conditions. Compared with normal culture conditions, the growth of *D. radiodurans* was inhibited when 34.2–170.9 mM NaCl or 200–1000 mM sorbitol was added to the TGY medium (Figure 1a,b). However, the biofilm-forming ability of *D. radiodurans* was induced under extreme environmental conditions compared with normal conditions. Under normal conditions, *D. radiodurans* could form mature biofilms after 96 h of incubation, while it formed equivalent biofilm biomass at 24 h or 48 h after the addition of 68.4–136.8 mM NaCl or 400–1000 mM sorbitol to TGY medium (Figure 2a,b). These results indicate that extreme osmotic stress can induce *D. radiodurans* biofilm formation.

Our previous studies demonstrated that both *carD* (*dr_2472*) and *dtrA/B* (*dr_A0009*/*dr_A0010*) play important regulatory roles in the resistance of *D. radiodurans* R1 to H_2_O_2_, UV radiation, and heat shock (Appendix A). To gain further insights into the potential roles of stress response-related genes in extreme biofilm formation, we measured the biofilm-forming capacities of the mutants devoid of *irrE*, *dtrA*, *dtrB*, *pprM*, or *carD* under osmotic stress conditions (136.8 mM NaCl and 800 mM sorbitol). Compared with the wild-type control, the deletion of *pprM* and *carD* resulted in decreased biofilm formation under osmotic stress conditions; the *carD* mutant, in particular, showed an almost total loss of biofilm production after 96 h of incubation when exposed to extreme concentrations of NaCl (Figure 3), indicating that both PprM and CarD are essential for the extreme biofilm formation of *D. radiodurans*.

### 2.2. Biofilms Are Beneficial to the Adaptation of D. radiodurans to Extreme Conditions

Biofilms are a key adaptation for microorganism survival in extreme environments, such as those involving heavy metal pollution, extreme temperature, high radiation, and hydrogen peroxide. To determine whether the biofilm formation of *D. radiodurans* plays a significant role in stress tolerance under severe conditions, we tested the survival rates of both biofilm cells and planktonic cells under H_2_O_2_ stress, a lethal dose of UV radiation, and sorbitol stress conditions.

For the H_2_O_2_ stress assay, after 80 mM of H_2_O_2_ treatment for 30 min, the survival rates were 36.02 ± 0.67% and 0.06 ± 0.04% for biofilms and planktonic cells, respectively. In addition, the abundance of surviving planktonic cells decreased by four orders of magnitude compared with the controls, but the abundance of surviving biofilm samples was reduced by zero to one order of magnitude. After treatment with 120 mM H_2_O_2_ and 160 mM H_2_O_2_, planktonic cells did not survive, but biofilm cells showed survival rates of 8.97 ± 0.28% and 7.11 ± 2.23%, respectively. At the same time, biofilm cells decreased by one order of magnitude in comparison with the controls without H_2_O_2_ treatment (Figure 4a).

For the UV radiation test, the results showed that both biofilms and planktonic cells of *D. radiodurans* had similar survival rates following non-UV fluence treatment. However, samples exposed to the highest UV fluence applied (160.97 kJ/m^2^) exhibited survival rates corresponding to 54.60 ± 3.02% and 0.93 ± 0.13% (for biofilms and planktonic samples; means ± SD) compared with the nontreated controls. Moreover, the results of drop plate methods showed that the abundance of planktonic cells decreased by approximately two orders of magnitude, while biofilm cells showed no difference (Figure 4b).

For the sorbitol stress experiment, the results showed that both biofilms and planktonic cells of *D. radiodurans* were able to survive under 3 M sorbitol stress. However, compared with the nontreated controls, the abundance of planktonic cells was reduced by two and three orders of magnitude after 7 h and 26 h of treatment, respectively, based on the drop plate results, while the biofilm cells showed no difference. Moreover, after 3 M sorbitol treatment for 26 h, the survival rate of planktonic samples was reduced dramatically to 0.08 ± 0.05%, and that of the biofilm samples was 30.20 ± 1.77% (Figure 4c).

The above results indicate that *D. radiodurans* biofilm cells were more stress-tolerant than planktonic cells under H_2_O_2_ stress, harsh UV irradiation, and sorbitol stress conditions. These findings support the hypothesis that biofilms contribute to the survival of *D. radiodurans* under extreme environmental conditions.

### 2.3. Genome-Wide Transcriptome Analyses of D. radiodurans Cells under Biofilm Formation Conditions

Using RNA-seq, the gene expression of *D. radiodurans* in four stages of biofilm development (the adhesion stage (T30), the microcolony formation stage (T50), the later-microcolony formation stage (T53), and the mature stage (T66)) was compared with expression in planktonic cells (initial stage (T0)). Nineteen differentially expressed genes were further examined using quantitative real-time PCR. As shown in Appendix A, the expression profiles of the selected genes showed the same tendencies as those detected with RNA-seq, indicating the good quality of the sequencing data. Primarily, a total of 409 differentially expressed genes (DEGs) (|log2(fold change)| > 1 and *p*-value < 0.05) were identified during *D. radiodurans* biofilm development (Appendix A). The analysis of KEGG pathways revealed that many DEGs are involved in biological processes that help cells adapt to the environment and biofilm formation, such as oxidative phosphorylation, endocrine resistance, quorum sensing, and glyoxylate and dicarboxylate metabolism. Of those genes, only 13 matched genes, accounting for 3.91% of 409 genes, share sequence homology with genes related to biofilm formation of *Escherichia coli*, *P. aeruginosa*, and *S. aureus*; however, based on a previous study, these homologous genes were not crucial transcription regulatory factors during biofilm development (Table 1).

To further identify the cellular pathways that were affected by biofilm development in *D. radiodurans*, the altered transcripts were classified according to the COG functional classification system, and the relative occurrence of genes belonging to each category is shown in Figure 5. Remarkably, unknown functional genes accounted for 64.30% of all DEGs, indicating the specificity of the regulatory network of *D. radiodurans*. Most intriguingly, twenty-nine genes associated with energy metabolism, carbohydrate metabolism, environmental signal processing, nucleotide metabolism, and stress response, as well as unknown functional genes, were shown to be differentially expressed during four stages of biofilm formation (Table 2), suggesting that these genes might play an essential role in the biofilm formation of *D. radiodurans*.

### 2.4. Identification of the Core drRRA Regulon Governing Biofilm Formation in D. radiodurans

It is widely known that TCSs play a critical role in bacterial adaptation and survival under various extreme environmental conditions. The response regulator DrRRA was previously identified to be essential for the extreme radioresistance of *D. radiodurans* [30]. Moreover, among 409 DEGs during WT biofilm development, at least 228 genes (Appendix A) are also affected by the *drRRA* mutation [30]. qRT–PCR analysis showed that no significant difference in *drRRA* expression during the biofilm formation process (Figure 6a), indicating that DrRRA is expressed constitutively during biofilm formation. To establish whether DrRRA participates in biofilm formation, we first ensured that there were no growth differences between the *drRRA* mutant strain, corresponding complemented strain, and wild-type strain in TGY medium (Figure 6b). Furthermore, the phenotype analysis results showed that the *drRRA* mutant could not form biofilms, while the complementary strain *drRRA-com* had restored biofilm formation ability (Figure 6c). We conclude that the response regulator DrRRA is essential for biofilm formation of the extremophilic bacterium *D. radiodurans* R1.

To further elucidate the effect of DrRRA on the biofilm-forming regulatory network of *D. radiodurans* R1, global transcriptional profiling analyses of the wild-type bacteria and the *drRRA* mutant were performed under the same conditions as were applied for the genome-wide analysis. Compared with the WT, *drRRA* deletion resulted in the expression levels of a total of 677 genes exhibiting more than a two-fold change across the several stages of biofilm formation (Figure 7a). In particular, 132 genes were found to be differentially expressed during the four stages of biofilm development, and the downregulated genes were assigned to COG terms, comprising four signal transduction, four energy production and conversion, three amino acid transport and metabolism, two nucleotide transport and metabolism, five carbohydrate transport and metabolism, one lipid transport and metabolism, two replication, recombination and repair, four cell wall/membrane/envelope biogenesis, five posttranslational modification, protein turnover, chaperones, seven inorganic ion transport and metabolism, and 88 function unknown (Figure 7b and Appendix A). Of these 132 differentially expressed genes, 125 genes (94.5%) were downregulated in Δ*drRRA* vs. the WT. Furthermore, only 2 of the 13 homology genes presented in Table 1, *dr_0408* and *dr_0888*, were differentially expressed in the *drRRA* mutant during the four stages of biofilm development. However, the DrRRA-binding site was only discovered in the promoter of *dr_0408*, indicating that DrRRA plays a direct regulatory role in the expression of *dr_0408*. Combined with biofilm formation by Δ*drRRA*, we inferred that DrRRA plays a positive regulatory role in the biofilm formation of *D. radiodurans*.

### 2.5. DrRRA Regulates Biofilm Formation via Transcriptional Activation of dr_0392

Based on the comparative transcriptome analysis of the WT and *drRRA* mutant under biofilm formation conditions, we found that the expression level of *dr_0392*, which encodes a membrane-binding protein, was significantly downregulated in the *drRRA* mutant (Appendix A). It has been reported that DR_0392 has the BON (bacterial OsmY and nodulation) domain and is involved in the resistance of *D. radiodurans* to NaCl stress and *γ*-irradiation [50]. We inferred that DR_0392, termed DrBON1 (for *Deinococcus radiodurans*
BON domain-containing protein 1), might play a role in biofilm formation under extreme environmental conditions. To test this hypothesis, the *drBON1* mutant (Δ*drBON1*) was constructed. While the *drBON1* mutant grew at a similar rate and to the same final optical density as the WT in the TGY medium, the mutant lost its biofilm formation capacity, especially under NaCl stress conditions (Figure 8a,b), indicating that DrBON1 was indeed involved in the biofilm formation of *D. radiodurans* under extreme NaCl concentration conditions.

The expression of drBON1 is associated with environmental stresses and might be regulated by various stress-related regulatory proteins, such as DrRRA, IrrE, DtrB, CarD, and DR_0891. dr_0891 encodes a response regulator in *D. radiodurans*, but our previous study showed that the biofilm formation ability of the dr_0891 mutant is similar to the WT. Thus, we compared the expression level of drBON1 between the wild type and strains with each of the six genes mutated. The qRT–PCR results showed that the expression of drBON1 was almost completely suppressed in the drRRA mutant (Figure 8c). Furthermore, we also found two putative DrRRA-binding sites (TCCTGAAAGCCCTG and TTCCTAAGAAGCTACGTC) in the promoter region of drBON1. The evidence available thus far strongly suggests that drBON1 is essential for biofilm formation under extreme NaCl concentration conditions and that its expression is dependent on DrRRA in a direct manner. To this end, we performed an electrophoretic mobility shift assay (EMSA) to verify the binding between DrRRA and the promoter of drBON1. The 6 × His-tagged DrRRA in the range of 0–8 μg was initially incubated with 50 ng of the labeled drBON1 (Figure 8d), in which DNA was completely bound with 8 μg of DrRRA. When 5 μg of unlabeled probes were added to the reaction, the overloaded DNA was competitive in forming a complex with DrRRA, resulting in a surplus of labeled DNA and the disappearance of the labeled DNA-DrRRA complex (Figure 8d). These results indicate that DrRRA is a positive regulator of biofilm formation and could directly activate the transcription of drBON1, which is the key biofilm formation-related gene in *D. radiodurans* under extremely high NaCl concentration conditions.

## 3. Discussion

Biofilms are recalcitrant to extreme environments and can protect microorganisms from ultraviolet (UV) radiation, extreme temperature, extreme pH, high salinity, high pressure, poor nutrient availability, antibiotics, etc., by acting as “protective clothing” [7]. In recent years, research on biofilms has mainly focused on biofilm-associated infections and strategies for combating microbial biofilms. However, these studies do not encompass possible regulators of pathways involved in extreme biofilm formation. In this study, we observed that extremely high NaCl or sorbitol concentration stresses could induce *D. radiodurans* to form mature biofilms quickly compared with normal incubation conditions (Figure 1 and Figure 2). In addition, we found that the biofilm-forming capabilities of the stress response-related genes *pprM* and *carD* mutants were defective under extreme NaCl and sorbitol stress conditions, indicating that both CarD and PprM are essential for the biofilm development of *D. radiodurans* under extreme environmental conditions (Figure 3). It was reported that CarD plays an important regulatory role in the resistance of *Mycobacterium tuberculosis* to oxidation, starvation, DNA damage, and infection [51]. It is widely known that (p)ppGpp is the signal molecular for cell response to stress, which contributes to the regulation of many aspects of microbial cell biology, like growth, adaptation, and biofilm [52]. Particularly, (p)ppGpp has reported involvement in the biofilm formation of Streptococci [53], *P. aeruginosa* [54], *Pseudomonas putida* KT2440 [55], and *Mycobacterium smegmatis* [56]. We suspect that (p)ppGpp may be the clue underlying CarD and biofilm formation in *D. radiodurans* R1. Interestingly, there was no difference in growth between the *carD* mutant and the wild type under osmotic stress conditions; however, the expression levels of the (p)ppGpp metabolism-related genes *dr_1838* (*relA*) and *dr_1631* (*relQ*) were decreased to various extents in the *carD* mutant compared with the wild type (Appendix A). Furthermore, the *relA* and *relQ* genes are involved in the response to oxidative, heat shock, and starvation stresses in *D. radiodurans* R1 [57]. Thus, we concluded that CarD may be involved in the extreme-biofilm formation of *D. radiodurans* through a particular osmotic stress response mechanism. PprM plays a role in the response to desiccation and oxidation stress [33,34]. There is no obvious direction for its role in biofilm formation. However, based on the TEM (Transmission Electron Microscope) observation results of our lab, the cell structure of the *pprM* mutant differs from the WT. This may be related to its biofilm formation phenotype. Furthermore, the regulatory mechanisms of CarD and PprM are still unclear and need to be clarified in the future. Further studies revealed that biofilms can increase cell resistance against UV irradiation, extreme oxidative stress, and sorbitol stress (Figure 4), indicating that biofilms are a good survival strategy for *D. radiodurans* under extreme environmental conditions. In *D. radiodurans*, the response regulator DrRRA plays a crucial regulatory role in biofilm formation, and *drBON1* is an essential gene involved in biofilm development, whose transcription is directly activated by DrRRA (Figure 6 and Figure 8).

Based on genome annotation and bioinformatics analysis, there are 25 response regulators in *D. radiodurans* R1 [58,59]. For the response regulator DrRRA (DR_2418), which was identified to be involved in biofilm formation regulation in this study, although there is a gene cluster, *dr2418-dr2419-dr2420,* located in the genome that encodes products composed of “RR-HK-RR”, the counterpart HK of the response regulator DrRRA has not been identified [30,60]. In addition, response regulator DR_0891 has no effect on biofilm development based on our previous study. Furthermore, the encoding genes of response regulators DR_0408 and DR_A0350 are DEGs during biofilm formation (Table 1). Moreover, DR_0987 (DqsR), a response regulator, is the only reported quorum sensing regulator in *D. radiodurans*, and there may be a cooperative relationship between DrRRA and DqsR when facing stress conditions [61]. Our results showed that *dqsR* was not differentially expressed at the transcriptional level in either the WT or the *drRRA* mutant. As components of TCS, response regulators, such as DrRRA and DqsR, play a role in signal transmission by phosphorylation; therefore, it is reasonable to observe no transcriptional changes in these two genes. However, whether DqsR participates in biofilm development warrants further exploration. And whether other regulators are involved in biofilm regulation circuits remains unknown. In general, the molecular regulatory mechanism of microorganism biofilm formation is complicated. In addition to TCS, quorum sensing, c-di-GMP, sigma factor, and small RNAs also play a role in microbe biofilm development [62,63,64,65,66,67]. Further studies are needed to explore the roles of these potential regulatory elements, such as those of small RNAs [68], in the *D. radiodurans* biofilm formation gene network.

The transcriptome results showed that only 13 DEGs were homologous to characterized biofilm-related genes in *E. coli*, *P. aeruginosa*, and *S. aureus* (Table 1), indicating the specific biofilm-forming regulatory network of *D. radiodurans*. Among the 13 DEGs encoding products of biofilm development, DR_0408 (32.74% for CheY and 30.47% for CpxR in *E. coli*) was downregulated in the *drRRA* mutant at all biofilm development stages compared with the WT strain (Appendix A). It is well-known that polysaccharides are the dominant component of biofilm EPS [3,69]. Moreover, there are DEGs in the *drRRA* mutant that participate in sugar metabolism, including *dr_1480*, *dr_A0047*, *dr_A0031*, and *dr_1689*. Among these, *dr_1480*, a member of the *dr_1473-dr_1483* gene cluster based on KEGG prediction, encodes alginate biosynthesis protein (AlgP); alginate is reported to be an important polysaccharide component in *P. aeruginosa* biofilm EPS [70]. Furthermore, DR_A0033*,* an ExoP-like protein, was involved in the exopolysaccharides synthesis of *D. radiodurans*, and compared with the WT, the biofilm biomass of the *dr_A0033* mutant is reduced [71]. However, its transcription is not differentially expressed during biofilm development in the WT, but compared with the WT, its expression was affected in the *drRRA* mutant (log2(fold change) value = 1.99 (*p* < 0.05) in T66/M66). Thus, the underlying mechanism of *dr_A0033* and DrRRA during biofilm formation could be explored in the future.

The proteins DrBON1 and DR_0888 of the extremophile *D. radiodurans* contain the BON (bacterial OsmY and nodulation) domain. The BON domain is supposed to be a membrane-binding domain that interacts with phospholipid membranes and is found in proteins such as the bacterial osmotic shock-resistance protein OsmY [72]. Moreover, it was reported that the transcription of *drBON1* and *dr_0888* was induced by NaCl stress and might play a role in the resistance of *D. radiodurans* to NaCl stress and *γ*-irradiation [50]. However, whether these two stress response-related genes are involved in biofilm formation is unknown. In this study, we discovered that *drBON1* participates in the biofilm development of *D. radiodurans* under normal growth and extreme environmental conditions (Figure 8), and *drBON1* transcription occurs in a DrRRA-dependent manner based on qRT–PCR analysis (Figure 8c) and transcriptome analysis (Appendix A). Although the expression of *dr_0888* was downregulated both at the microcolony formation stage (T50) (Table 1) and in the *drRRA* mutant (Appendix A), the putative DrRRA-binding site was not found in the promoter region of *dr_0888*.

## 4. Materials and Methods

### 4.1. Bacterial Strains, Plasmids, and Growth Conditions

All bacterial strains and plasmids used in this study are described in Appendix A. *D. radiodurans* (CGMCC 1.633) was obtained from the China General Microbiological Culture Collection Center (Beijing, China). *D. radiodurans* and its derivatives were cultured in liquid TGY medium (1.0% tryptone, 0.1% glucose, and 0.5% yeast extract) at 30 °C or cultured on solid TGY medium with 1.5% agar. When needed, kanamycin, spectinomycin, and chloromycetin were added to final concentrations of 20 µg/mL, 350 µg/mL, and 3.4 µg/mL, respectively. *E. coli* strains were cultured in LB medium (1.0% tryptone, 1.0% NaCl, and 0.5% yeast extract).

### 4.2. Construction of the Mutants and Complementary Strains

The Δ*drBON1* (Δ*dr_0392*) deletion mutant was constructed with double crossover recombination of a spectinomycin (spec) resistance cassette into the genome, as described previously [73]. PCR primers were designed based on the sequence of *drBON1* in the genome. First, *dr_0392*-Up (828 bp), *dr_0392*-Sp (946 bp), and *dr_0392*-Down (810 bp) were amplified. Then, the three amplified DNA fragments were used as templates for the overlap reaction (at a ratio of 1:1:1), and the resulting PCR fragment (2512 bp) was directly transformed into *D. radiodurans*, as previously reported [74]. A single colony with spectinomycin resistance was selected. Finally, PCR and DNA sequencing were used to verify that the expression of the *drBON1* gene was knocked out. The successfully constructed mutant was named ∆*drBON1(*∆*dr_0392)*. For the complementation of the *drRRA* gene in *D. radiodurans*, *drRRA* and its promoter were inserted into pRADZ3 using a Seamless Assembly Mix, and then the recombinant plasmid was transformed into the *drRRA* mutant. The primers were synthesized by Shanghai Sangon Biotech (Sangon Biotech, Shanghai, China), and the sequences are shown in Appendix A.

### 4.3. Biofilm Formation Assays under Normal, NaCl, and Sorbitol Conditions

Surface-adhered biofilm formation was examined using the crystal violet (CV) method and was performed in 96-well microtiter plates. *D. radiodurans* used for biofilm experiments was grown in TGY at 30 °C with shaking at 220 rpm to an OD_600_ = 10–12. Cultures were harvested with centrifugation at 5000× *g* for 5 min and washed twice in fresh TGY broth. Cultures were resuspended in fresh TGY broth (supplemented with different concentrations of NaCl (0–170.9 mM) or sorbitol (0–1000 mM), if needed) and adjusted to OD_600_ = 0.1 ± 0.03 in fresh TGY medium. A total of 150 μL of culture was aliquoted into separate wells in a 96-well PVC plate. Microtiter plates were carefully wrapped using sterilized parafilm and placed in a 30 °C incubator without agitation for 12 h, 48 h, 72 h and 96 h. At the indicated time points, nonadherent planktonic cells were removed using a multichannel pipette without disturbing the biofilm area, and individual wells were washed twice with 160 μL of sterile double-distilled H_2_O. Then, 160 μL of 1% (*w*/*v*) CV solution in ethanol was added to each well for 20 min and washed four times with 200 μL of ddH_2_O. Photos were taken, and the cell-associated CV was solubilized with 30% acetic acid and quantified by measuring the A595 of the resulting solution using FlexStation 3 (Molecular Devices, Sunnyvale, CA, USA). Statistical analysis was performed in Microsoft Excel 2021 using one-way analysis of variance (ANOVA).

### 4.4. RNA Isolation and Quantitative Real-Time PCR (qRT–PCR)

Total RNA was isolated with TRIzol reagent and a PureLink RNA Mini kit (Invitrogen, Carlsbad, CA, USA) following the manufacturer’s instructions. RNA purity was assessed using a NanoPhotometer^®^ NP80 (Implen, Munich, Germany). RNA (1 µg) was reverse transcribed to cDNA using a PrimeScript^TM^ RT reagent kit with gDNA Eraser (TaKaRa Bio, Takara, Japan) as described in the manufacturer’s protocol. qRT–PCR was performed with cDNA (1.0 µL) obtained from three independent cultures using ChamQ SYBR qPCR Master Mix (Vazyme Biotech, Nanjing, China) on a 7500 Fast Real-Time PCR System (Applied Biosystems, Foster City, CA, USA). Primers were designed based on the full genome sequence of *D. radiodurans* R1 (accession number: CP015081 or AE000513), and they are listed in Appendix A. The 16S rRNA gene was used as the endogenous reference gene to normalize the expression of target genes in each cDNA template. The amplification conditions of the qRT–PCR assays were as follows: 95 °C for 5 min, followed by 40 cycles of 95 °C for 30 s, 60 °C for 1 min, and 72 °C for 30 s and then a dissociation curve analysis. The relative expression level of target genes was calculated using the comparative threshold cycle (2^−∆∆Ct^) method [75]. The qRT–PCR assays were performed using total RNA preparations obtained from three independent cultures.

### 4.5. RNA Sequencing and Analysis

Bacterial cell cultures with OD_600_ = 0.1 ± 0.03 in fresh TGY medium were prepared as described in Section 4.3. Then, 1.5 mL of the culture was transferred to sterilized 24-well plates (Costar^®^ 3524) (Corning Costar, Kennebunk, ME, USA). Microtiter plates were carefully wrapped using sterilized parafilm and placed in a 30 °C incubator without agitation. Four stages of biofilm growth were determined based on pretest observations for the biofilm formation stages in a 96-well PVC plate and 24-well plate at different time periods. At the indicated time points, 1.5 h (as sample T0 or M0), 30 h, 50 h, 53 h, and 66 h, all cells were harvested with centrifugation at 12,000× *g* for 3 min and stored at −80 °C. All samples were of three biological replicates. The biofilm phenotype of these five periods was observed in a duplicate sample plate. Total RNA was extracted using TRIzol^®^ LS reagent (Invitrogen, Carlsbad, CA, USA) following the manufacturer’s instructions. Bacterial 23S and 16S rRNAs were subsequently depleted with a MICROBExpress bacterial mRNA enrichment kit (Ambion, Austin, TX, USA). Fragmentation buffer was added to obtain a fractioned mRNA template. First strand cDNA was synthesized using random hexamers as primers, followed by second-strand cDNA synthesis using DNA Polymerase I and RNase H. After purification with AMPure XP beads, the cDNA fragments were end-repaired with the addition of a single “A” base at the 3′-end of each strand and subsequently ligated to special sequencing adapters. Then, total RNA-seq libraries were constructed using an Truseq RNA sample prep Kit (Illumina Inc., San Diego, CA, USA) with PCR and sequenced using an Illumina HiSeq 2000 instrument and the paired-end method by Tianjin Biochip Corporation (Tianjin, China).

Clean reads were achieved after moving the adaptor, low-quality reads, and length < 40 nt reads of the raw reads. Then, the clean reads were mapped to the reference genome of *D. radiodurans* R1 (accession number: CP015081). Bowtie2 [76] (with the mismatch value set as 2 and the other parameters set as default) was used to build the reference genome index, and then, the FPKMs of the coding genes in each sample were calculated using Cuffdiff [77]. Genes with |log2(fold change)| > 1 and *p* < 0.05 were considered DEGs. The quality of all steps was controlled in accordance with the recommendations of Illumina.

### 4.6. Cell Survival under Oxidative Stress, Sorbitol Stress, and UV Radiation

To test the resistance of biofilm or planktonic cells to oxidative and osmotic stresses, the biofilm sample was washed twice with sterile phosphate-buffered saline (PBS, 0.02% KH_2_PO_4_, 0.29%Na_2_HPO_4_·12H_2_O, 0.8% NaCl, 0.02% KCl, pH = 7.5), and then, both the biofilm and planktonic cells (OD_600_ = 0.6–0.8) were treated with different concentrations of H_2_O_2_ or 3 M sorbitol for the indicated time. After treatment, the biofilm cells were washed twice with sterile PBS buffer, harvested by scraping, and resuspended in 200 µL PBS buffer. To test biofilm cell resistance to UV radiation, 50 µL of culture (OD_600_ = 0.1 ± 0.03) was inoculated on black polycarbonate filter membranes placed in a TGY plate, and the plates were put into a 30 °C incubator for 96 h. Subsequently, the whole membranes were transferred into sterilized petri dishes and treated with different doses of 254 nm UV(UV-C) radiation. The irradiation dose was estimated with a UV radiometer (Photoelectric Instrument Factory, Beijing Normal University) using a detector at 254 nm. Then, the biofilm cells were collected in 500 µL of sterile PBS buffer. For the planktonic cell analysis under UV-C radiation, the fresh culture (OD_600_ = 7–9) was harvested with centrifugation (5000× *g*, 5 min), washed twice with sterile PBS buffer, and adjusted to OD_600_ = 8.0. Then, the cell culture (500 µL) dropped on the black polycarbonate filter membranes was treated with UV radiation. For all survival analyses, after stress treatment, serial 10-fold dilutions of these cells (8 µL) were spotted on TGY plates and incubated at 30 °C. Furthermore, 100 µL of the dilution samples were spread on TGY agar plates to calculate the number of colony-forming units (CFUs). The initial cell numbers of the untreated samples were within the same order of magnitude for both biofilms and planktonic samples. The survival rate was expressed as the percentage of the number of colonies in the treated samples compared with that in the untreated sample [19]. These experiments included three independent biological replicates. The assays were repeated with two biological replicates. Statistical analysis was performed using one-way analysis of variance (ANOVA) in Microsoft Excel 2021.

### 4.7. Electrophoretic Mobility Shift Assay (EMSA)

His-tagged DrRRA was expressed using the pET28a plasmid in *E. coli* BL21(DE3) and purified with Ni-NTA magnetic agarose beads (QIAGEN, Hilden, Germany). For the preparation of fluorescent FAM-labeled probes, the promoter region of ∆*drBON1* was PCR-amplified with Dpx DNA polymerase (TOLO Biotech, Shanghai, China) from the recombinant plasmid pJET1.2 using primers pJET1.2F and pJET1.2R (FAM). The FAM-labeled probes were purified using the Wizard^®^ SV Gel and PCR Clean-Up System (Promega, Madison, WI, USA) and quantified with a NanoDrop 2000C (Thermo Fisher Scientific, Waltham, MA, USA). The EMSA reaction sample (20 µL) contained a 50 ng labeled probe and different quantities of DrRRA protein in a reaction buffer of 50 mM Tris-HCl (pH = 8.0), 100 mM KCl, 2.5 mM MgCl_2_, 0.2 mM DTT, 2 μg salmon sperm DNA, and 10% glycerol. After incubation at 25 °C for 30 min, the reaction system was loaded into 6% PAGE buffered with 0.5× TBE. Gels were scanned with ImageQuant LAS 4000 mini (GE Healthcare, Chicago, IL, USA).

## Figures and Tables

**Figure 1 ijms-25-00421-f001:**
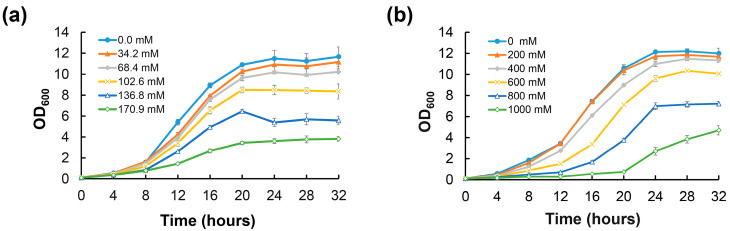
Effect of NaCl concentration (0–170.9 mM) (**a**) and sorbitol concentration (0–1000 mM) (**b**) on the growth of *D. radiodurans*. Growth was monitored as the absorbance at 600 nm. The mean values of the results from three experiments are shown, and error bars represent the standard deviation of three biological replicates.

**Figure 2 ijms-25-00421-f002:**
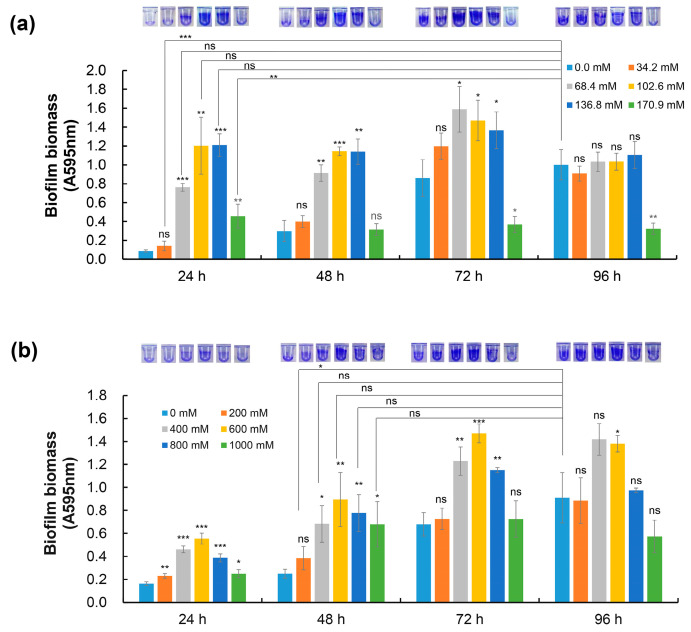
Effect of NaCl concentration (0–170.9 mM) (**a**) and sorbitol concentration (0–1000 mM) (**b**) on biofilm biomass using the crystal violet (CV) method. CV staining of the biofilm obtained is shown at the top. Error bars represent the standard deviation of three biological replicates. Asterisks indicate statistical significance determined using one-way ANOVA: *** *p* < 0.001, ** *p* < 0.01, * *p* < 0.05, and ns: non-significant. Unless indicated by lines, the *p*-values were calculated between samples and the 0.0 mM NaCl or 0 mM sorbitol control sample at the same time points.

**Figure 3 ijms-25-00421-f003:**
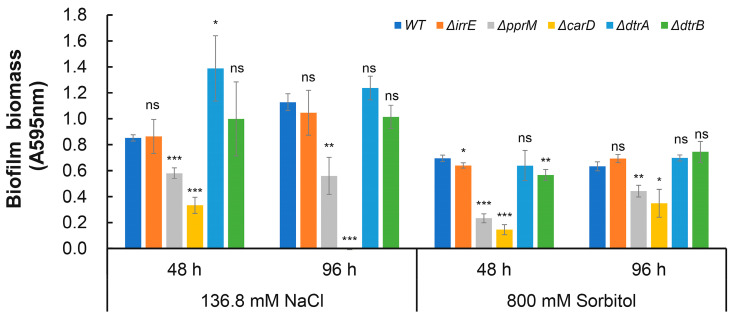
Effect of abiotic stress response gene deletion on biofilm formation 48 h and 96 h after inoculation with a comparison of biofilm biomass obtained from the WT under 136.8 mM NaCl conditions and 800 mM sorbitol conditions. Error bars represent the standard deviation of three biological replicates. Asterisks indicate statistical significance determined using one-way ANOVA: *** *p* < 0.001, ** *p* < 0.01, * *p* < 0.05, and ns: nonsignificant.

**Figure 4 ijms-25-00421-f004:**
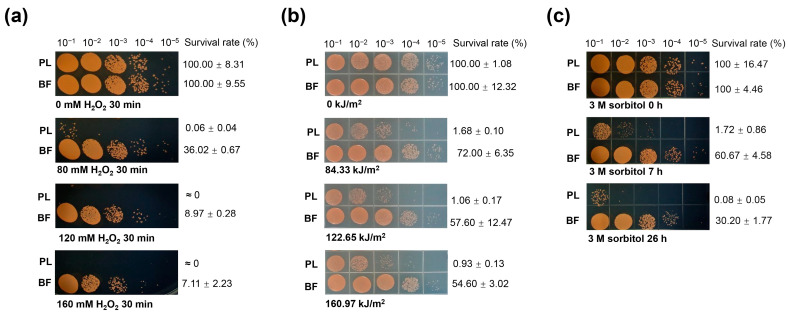
Survival assays for planktonic and biofilm cells of *D. radiodurans* R1 following exposure to H_2_O_2_ treatment (**a**), UV radiation, (**b**) and sorbitol stress (**c**). BF, biofilm cell samples; PL, planktonic cell samples.

**Figure 5 ijms-25-00421-f005:**
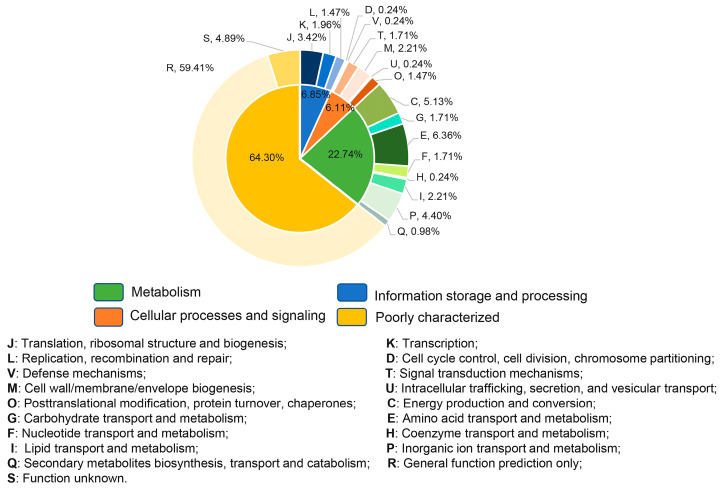
Analysis of differentially expressed genes in *D. radiodurans* R1 under biofilm formation conditions. Functional categories of genes differentially regulated in *D. radiodurans* R1 under biofilm formation conditions by Clusters of Orthologous Groups (COGs) of proteins.

**Figure 6 ijms-25-00421-f006:**
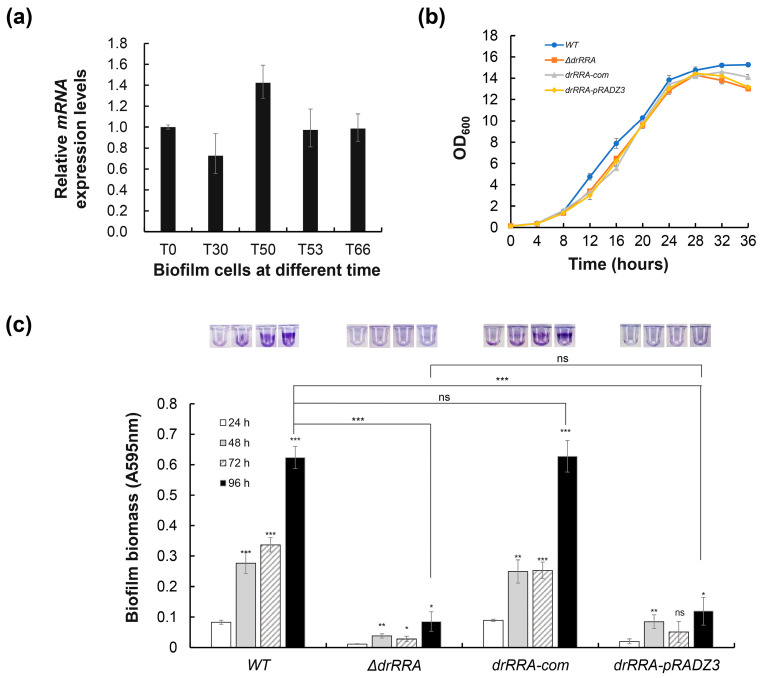
Expression and regulatory role analysis of *drRRA* during biofilm formation. qRT–PCR analysis of *drRRA* expression in different biofilm formation stages (**a**). Growth curve (**b**) and biofilm biomass assay (**c**) of the wild type, the *drRRA* mutant, complementary strain *drRRA-com*, and recombinant strain *drRRA*-pRADZ3 under normal TGY growth conditions. Error bars represent the standard deviation of three replicates. Asterisks indicate statistical significance determined using one-way ANOVA: *** *p* < 0.001, ** *p* ≤ 0.01, * *p* < 0.05, and ns: nonsignificant.

**Figure 7 ijms-25-00421-f007:**
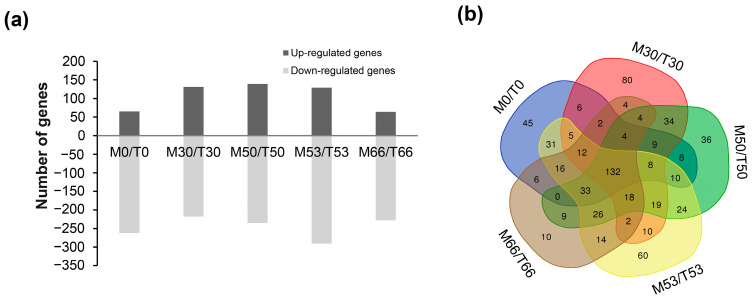
The statistical figure (**a**) and Venn diagram (**b**) of differentially expressed genes in the mutant strain Δ*drRRA* under biofilm formation conditions at different stages.

**Figure 8 ijms-25-00421-f008:**
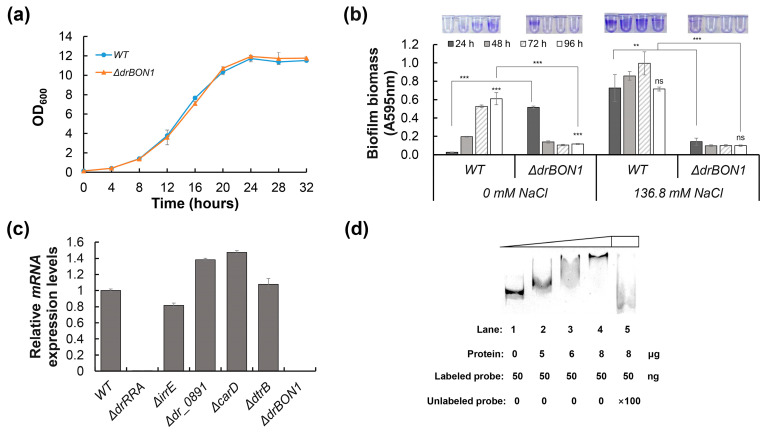
The biofilm-related gene *drBON1* is transcriptionally regulated by DrRRA in *D. radiodurans*. Growth curve (**a**) and biofilm formation ability assay (**b**) of the wild type and Δ*drBON1* mutant under normal growth conditions. (**c**) Transcriptional analysis of *drBON1* in *D. radiodurans* R1 and related mutants using qRT–PCR. (**d**) Electrophoretic mobility shift assay of DrRRA binding to the FAM-labeled *drBON1* promoter. Error bars represent the standard deviation of three replicates. Asterisks indicate statistical significance determined using one-way ANOVA: *** *p* < 0.001, ** *p* < 0.01. and ns: nonsignificant.

**Table 1 ijms-25-00421-t001:** Homology analysis of differentially expressed genes in *D. radiodurans* R1.

Locus Tag	Annotation	Homology Protein	Identity (%)	Coverage (%)	Log2(Fold Change)			
					T30/T0	T50/T0	T53/T0	T66/T0
DR_0408	response regulator	CheY (NP_416396.1; *E. coli*) [37]	32.74	69	−0.24	**−1.40**	**−1.09**	**−1.28**
		CpxR (NP_418348.1; *E. coli*) [38]	30.47	49				
DR_A0350	response regulator PilH	CpxR (NP_418348.1; *E. coli*) [38]	30.25	49	**2.16**	**2.16**	**1.10**	**1.01**
DR_0997	Crp/Fnr family transcriptional regulator	Vfr (NP_249343.1; *P. aeruginosa*) [39]	20.92	91	**−2.15**	**-2.15**	−0.93	−0.82
DR_B0067	salicylate monooxygenase-related protein	PqsH (NP_251277.1; *P. aeruginosa*) [40]	34.41	23	−0.70	**−4.43**	**−4.88**	**−2.26**
DR_B0072	salicylate monooxygenase-related protein	PqsH (NP_251277.1; *P. aeruginosa*) [40]	40.79	18	−1.00	**−4.43**	**−4.78**	**−2.20**
DR_1198	GTP-binding elongation factor family protein TypA/BipA	TypA (NP_253804.1; *P. aeruginosa*) [41]	56.83	98	−0.61	**−1.72**	−0.77	−0.84
DR_1291	D-3-phosphoglycerate dehydrogenase	LdhA (NP_249618.1; *P. aeruginosa*) [42]	30.88	85	**1.54**	**1.79**	**1.72**	**2.08**
DR_0713	lipopolysaccharide glycosyltransferase, putative	WbpX (NP_254136.1; *P. aeruginosa*) [43]	16.38	61	−0.68	**1.37**	**2.03**	**2.30**
DR_1516	acetolactate synthase	AlsS (WP_000130155.1; *S. aureus*) [44]	29.4	96	**1.47**	**1.58**	**2.28**	**2.17**
DR_1998	catalase	KatA (NP_252926.1; *P. aeruginosa*) [45]	42.8	99	**2.38**	**2.69**	**2.30**	**2.17**
DR_0888	hypothetical protein	Spa (AFD54305.1; *S. aureus*) [46]	35.29	10	−0.66	**−1.45**	−0.55	−0.83
DR_0662	hypothetical protein	MazF (BBJ19047.1; *S. aureus*) [47]	26.79	93	−0.31	**−1.38**	**−1.17**	−0.79
DR_0687	hypothetical protein	Wza (NP_416566.1; *E. coli*) [48,49]	26.17	27	**−2.96**	−0.34	−0.22	0.03

The values of |log2(fold change)| > 1 are shown in bold font.

**Table 2 ijms-25-00421-t002:** Transcriptional characteristics of 29 genes that were differentially regulated at four distinct biofilm formation stages of *D. radiodurans* R1.

Locus Tag	Functional Description	Log2(Fold Change)			
		T30/T0	T50/T0	T53/T0	T66/T0
Energy metabolism					
DR_1501	NADH-quinone oxidoreductase	2.58	2.89	2.73	2.28
DR_1498	NADH-quinone oxidoreductase subunit H	2.92	2.59	3.62	2.89
DR_1497	NADH-quinone oxidoreductase subunit I	3.03	2.67	3.65	2.85
DR_1496	NADH dehydrogenase subunit J	2.33	2.46	3.30	2.64
DR_1495	NADH-quinone oxidoreductase subunit K	2.76	2.77	3.82	3.20
DR_1493	NADH dehydrogenase subunit M	2.76	2.76	3.72	3.02
DR_1492	NADH-quinone oxidoreductase subunit N	2.31	2.35	3.19	2.67
DR_B0106	cytochrome C oxidase subunit II	4.59	3.60	2.22	2.18
DR_1440	ATPase	3.06	3.55	4.24	3.63
Carbohydrate metabolism					
DR_0828	isocitrate lyase	4.74	4.57	6.13	4.72
DR_1778	3-isopropylmalate dehydratase large subunit 2	1.92	1.94	2.13	2.00
Environmental information processing					
DR_B0073	PTS fructose transporter subunit IIBC	−1.41	−3.92	−5.02	−2.06
DR_B0074	PTS fructose transporter subunit IIA	−1.39	−3.59	−4.38	−1.63
DR_B0075	1-phosphofructokinase	−1.69	−4.22	−4.57	−1.77
DR_1220	iron transporter	−2.67	−2.65	−1.72	−1.57
Nucleotide metabolism					
DR_B0107	ribonucleotide reductase, NrdI family	2.85	2.20	2.04	2.04
DR_B0108	ribonucleotide-diphosphate reductase subunit alpha	2.83	2.16	1.80	1.85
DR_B0109	ribonucleotide-diphosphate reductase subunit beta	2.86	1.93	1.45	1.55
DR_B0121	Iron ABC transporter, ATP-binding protein	−1.83	−3.24	−2.73	−2.79
DR_B0122	iron ABC transporter permease	−2.03	−3.30	−2.70	−2.80
DR_B0123	iron ABC transporter permease	−2.34	−3.10	−2.73	−2.61
Stress response					
DR_1998	catalase	2.38	2.69	2.29	2.17
Unknown function					
DR_1710	hypothetical protein	2.69	3.36	2.97	2.34
DR_1082	hypothetical protein	2.34	2.42	1.67	1.91
rna4	tRNA-Phe	ND *	ND *	ND *	ND *
rna8	tRNA-Glu	ND *	2.18	ND *	1.71
rna26	tRNA-Ser	ND *	1.62	ND *	ND *
rna54	tRNA-Ala	ND *	2.07	2.37	2.55
DR_2028	DNA-binding protein	ND *	−2.16	−3.07	−1.65

ND means no data. * There is no FPKM value in the corresponding biofilm formation stages sample (T30 or T50 or T53 or T66); therefore, the gene expression is downregulated.

## Data Availability

The data presented in this study are openly available in NCBI, accession numbers SAMN26029339, SAMN26029340, SAMN26029341, SAMN26029342, SAMN26029343, SAMN26029344, SAMN26029345, SAMN26029346, SAMN26029347, and SAMN26029348.

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
