# Peer review of "Development and Regulation of the Extreme Biofilm Formation of Deinococcus radiodurans R1 under Extreme Environmental Conditions"

_ijms, 2023, doi:10.3390/ijms25010421_

Round 1

Reviewer 1 Report

Comments and Suggestions for Authors

The study primarily focuses on the biofilm formation in D. radiodurans and its genetic and molecular aspects. This article includes the analysis of biofilm formation under osmotic stress, the role of specific genes in biofilm formation, and the response of these biofilms to different stress conditions (like H2O2, UV, and sorbitol). The paper includes information on the transcriptional characteristics of genes at different stages of biofilm formation in D. radiodurans. The article contributes new insights into the genetic regulation of biofilm formation in D. radiodurans, particularly in response to environmental stresses. Yet, several aspects need clarification before publication.

1. In Section 2.1, the authors argued that osmotic stress can induce D. radiodurans biofilm formation. Osmotic stress leads to both biofilm formation (Fig. 2) and growth retardation (Fig. 1) of D. radiodurans. It's crucial to determine whether osmotic stress directly causes increased biofilm formation or if biofilm formation is an indirect result of the stress-induced changes (i.e., growth arrest) in the cells. Analyzing the expression of genes involved in biofilm formation versus those involved in stress response and growth can help differentiate the effects of osmotic stress on these two processes. (1) The biofilm formation was assessed in various mutants (Fig. 3). However, the manuscript lacks detailed information on the specific genes investigated, particularly dtrAB and carD. While these genes are mentioned as stress response-related (lines 72-73), the explanation provided is insufficient. I would like to understand the rationale behind selecting these particular mutants for the study. Are they associated explicitly with osmotic stress response? It would benefit the readers if relevant references were cited to support the discussion of these genes. (2) The findings on the pprM and carD mutant's reduced biofilm-forming ability under high NaCl and sorbitol stress conditions suggest that both genes play a critical role in D. radiodurans' biofilm development in osmotically challenging environments. However, the paper currently does not offer an explanation or a detailed description of the two genes, nor does it elucidate the specific role of the genes in this context. Rather than stating that "the regulatory mechanism of PprM and CarD is still unclear and needs to be clarified in the future (lines 301-302)", it would be more beneficial for the authors to undertake additional experiments to investigate and elucidate this mechanism. A more thorough discussion and experimental evidence in this area would significantly enhance the paper's contribution to our understanding of the genetic regulation of biofilm formation in response to osmotic stress.

2. Section 2.3 focuses on genome-wide transcriptome analyses of D. radiodurans cells across four stages of biofilm development. (1) The rationale for selecting specific time points (30 h, 50 h, 53 h, and 66 h for the adhesion, microcolony formation, later-microcolony formation, and mature stages) is unclear. The authors must provide a clear basis or justification for choosing these particular time points to strengthen the study's methodology. (2) While the paper identifies 409 differentially expressed genes (DEGs), only 29 are listed in Table 2. The authors should include all 409 DEGs in a supplementary table to provide a comprehensive view. Furthermore, a detailed discussion on how these DEGs enable D. radiodurans to adapt to challenging conditions, as indicated in Figure 4, would significantly enrich the paper. (3) Table 1 lists 13 genes that share sequence homology with genes known to be involved in biofilm formation. However, there is a notable absence of discussion or explanation regarding the roles these genes may play in the biofilm-forming regulatory network of D. radiodurans. The study must investigate whether any of these genes influence biofilm development. For instance, the gene dr_0408 was observed to be downregulated in the drRRA mutant compared to the wild-type strain (lines 325–326). The role and impact of DR_0408 in biofilm formation should be examined and elaborated upon in the manuscript.

3. Section 2.4 elucidates the role of DrRRA in D. radiodurans' biofilm formation. I find the rationale for focusing on this gene to be insufficiently substantiated. While the manuscript acknowledges the importance of two-component systems (TCSs) in bacterial adaptation and survival under harsh conditions and identifies the response regulator DrRRA as a key factor in D. radiodurans' radioresistance (lines 205-207), this alone does not adequately justify its selection for study. It would seem more logical to examine DR_0997 (DdrI), as listed in Table 1. This gene is a homolog of one involved in biofilm formation and is a transcription regulator linked to D. radiodurans' stress response (DOI:10.1371/journal.pone.0155010). Why didn't the authors investigate DdrI?

A comparative analysis of the 409 DEGs during biofilm formation with those affected by drRRA mutation (DOI: 10.1111/j.1365-2958.2008.06113.x) would be valuable. If there is an overlap between these genes, it could provide a more compelling reason for selecting DrRRA. Additionally, the target gene drBON1 choice also appears to have a weak rationale. The paper notes that the expression level of dr_0392, a gene encoding a membrane-binding protein, was significantly downregulated in the drRRA mutant (lines 245-247). However, the expression levels of dr_1314 and dr_1199 were reduced even more (Table S2). Notably, dr_1314 is part of a heat shock regulon (DOI: 10.1128/JB.187.10.3339-3351.2005). Hence, it would be prudent for the authors to investigate the effects of other DrRRA-regulated genes on biofilm development to provide a more robust and comprehensive understanding of the underlying mechanisms.

The manuscript currently presents three distinct topics: (i) the induction of biofilm formation by osmotic stress, (ii) transcriptome analysis during biofilm development under normal conditions, and (iii) the effect of DrRRA on biofilm formation, but fails to integrate them cohesively. I recommend thoroughly revising the Discussion section to create a clear narrative thread that connects these topics, highlighting how each contributes to our understanding of D. radiodurans biofilm formation. Furthermore, the manuscript should be revised to consider the previously reported composition of D. radiodurans' EPS (doi.org/10.1038/s41598-019-56141-3). Specifically, lines 327–332 should be updated to reflect the current study's results and interpretations.

Minor comments

- line 66: Deinococcus metalilites à Deinococcus metallilatus

- line 76: replace '31' and '32' with more relevant references

- line 88: abiotic stress resistance à abiotic stress

- line 208: thatno à that no

- Fig. 8C: what is the dr_0891?

- line 266: delete the reference '30'

- line 299: card à carD

- line 329: dr_a0047 à dr_A0047

- line 375: correct ‘220 r/min’ and ‘OD600’

- line 430: specifiy the kinds of UV rays (UVA, B or C?) and the device used.

Author Response

Reviewer1:

Comments and Suggestions for Authors

The study primarily focuses on the biofilm formation in D. radiodurans and its genetic and molecular aspects. This article includes the analysis of biofilm formation under osmotic stress, the role of specific genes in biofilm formation, and the response of these biofilms to different stress conditions (like H2O2, UV, and sorbitol). The paper includes information on the transcriptional characteristics of genes at different stages of biofilm formation in D. radiodurans. The article contributes new insights into the genetic regulation of biofilm formation in D. radiodurans, particularly in response to environmental stresses. Yet, several aspects need clarification before publication.

In Section 2.1, the authors argued that osmotic stress can induce D. radiodurans biofilm formation. Osmotic stress leads to both biofilm formation (Fig. 2) and growth retardation (Fig. 1) of D. radiodurans. It's crucial to determine whether osmotic stress directly causes increased biofilm formation or if biofilm formation is an indirect result of the stress-induced changes (i.e., growth arrest) in the cells. Analyzing the expression of genes involved in biofilm formation versus those involved in stress response and growth can help differentiate the effects of osmotic stress on these two processes. The biofilm formation was assessed in various mutants (Fig. 3). However, the manuscript lacks detailed information on the specific genes investigated, particularly dtrAB and carD. While these genes are mentioned as stress response-related (lines 72-73), the explanation provided is insufficient. I would like to understand the rationale behind selecting these particular mutants for the study. Are they associated explicitly with osmotic stress response? It would benefit the readers if relevant references were cited to support the discussion of these genes.

A: Thank you for pointing this issue out. Both CarD and DtrA/B are key regulators controlling the stress resistance of D. radiodurans R1. It was reported that CarD plays an important regulatory role in the resistance of Mycobacterium to oxidation, starvation, DNA damage, and infection (Stallings CL et al., 2009, CarD is an essential regulator of rRNA transcription required for Mycobacterium tuberculosis persistence. Cell, 138(1):146-59). Our previous studies have demonstrated that extreme environmental conditions, such as strong UV radiation and H2O2, can stimulate carD expression in D. radiodurans R1, and the stress resistance of D. radiodurans R1 was impacted by the deletion of carD (unpublished data). Furthermore, bioinformatics analysis showed that DtrA/B is a two-component system, and a lack of the dtrA/B gene caused sensitivity to heat shock (unpublished data). These unpublished data showed that the three genes are involved in the regulation of D. radiodurans R1 extreme resistance. So, we wonder whether these genes are involved in the regulation of biofilm formation in D. radiodurans R1 under extreme stress environmental conditions. To address this comment, we supplemented the detailed information on the three genes (lines 78-80) as follows: Our previous studies showed that CarD and DtrA/B play important regulatory roles in the resistance of D. radiodurans R1 to H2O2, high UV radiation or heat shock (unpublished data).

The findings on the pprM and carD mutant's reduced biofilm-forming ability under high NaCl and sorbitol stress conditions suggest that both genes play a critical role in D. radiodurans' biofilm development in osmotically challenging environments. However, the paper currently does not offer an explanation or a detailed description of the two genes, nor does it elucidate the specific role of the genes in this context. Rather than stating that "the regulatory mechanism of PprM and CarD is still unclear and needs to be clarified in the future (lines 301-302)", it would be more beneficial for the authors to undertake additional experiments to investigate and elucidate this mechanism. A more thorough discussion and experimental evidence in this area would significantly enhance the paper's contribution to our understanding of the genetic regulation of biofilm formation in response to osmotic stress.

A: PprM is a radiation stress response protein, which deletion increases the sensitivity of D. radiodurans to γ-rays and UV and also regulates the catalase KatE1, the synthesis of deinoxanthin and the concentration of metal ions in D. radiodurans (Jeong SW et al., 2016, PprM is necessary for up-regulation of katE1, encoding the major catalase of Deinococcus radiodurans, under unstressed culture conditions. J Microbiol, 54(6):426-31; Zeng Y et al., 2017, Knockout of pprM decreases resistance to desiccation and oxidation in Deinococcus radiodurans. Indian J Microbiol, 57(3):316-321). carD, a widely distributed DNA damage and starvation inducible gene, whose protein product binds the RNAP to control rRNA transcription in mycobacteria, plays an important regulatory role in the resistance of Mycobacterium to oxidation, starvation, DNA damage, and infection (Stallings CL et al., 2009, CarD is an essential regulator of rRNA transcription required for Mycobacterium tuberculosis persistence. Cell, 138(1):146-59). These findings showed that both PprM and CarD are key stress response-related regulators. In this study, we found that the deletion of pprM and carD resulted in decreased biofilm formation of D. radiodurans R1, indicating that the two genes also play an important regulatory role in biofilm formation under osmotic stress conditions, but the underlying mechanisms are unclear and need to be clarified in future. To address this comment, we supplemented the detailed information on PprM and CarD (lines 76-80) and modified the discussion (lines 314-330).

Section 2.3 focuses on genome-wide transcriptome analyses of D. radiodurans cells across four stages of biofilm development. The rationale for selecting specific time points (30 h, 50 h, 53 h, and 66 h for the adhesion, microcolony formation, later-microcolony formation, and mature stages) is unclear. The authors must provide a clear basis or justification for choosing these particular time points to strengthen the study's methodology.

A: In order to explore the biofilm formation characteristics of D. radiodurans R1, its biofilm formation ability at different time periods was detected by using the crystal violet (CV) method. On the basis of pretesting, four stages of biofilm development were determined: the adhesion stage (T30), the microcolony formation stage (T50), the later-microcolony formation stage (T53) and the mature stage (T66)). To address this comment, the former section of RNA Sequencing was substantially revised (lines 454-456).

While the paper identifies 409 differentially expressed genes (DEGs), only 29 are listed in Table 2. The authors should include all 409 DEGs in a supplementary table to provide a comprehensive view. Furthermore, a detailed discussion on how these DEGs enable D. radiodurans R1 to adapt to challenging conditions, as indicated in Figure 4, would significantly enrich the paper.

A: Thank you for pointing this issue out. A new Table S5 of the 409 differentially expressed genes (DEGs) was added in the Supplementary Materials. Furthermore, a detailed discussion on how these DEGs enable D. radiodurans R1 to adapt to challenging conditions based on KEGG pathway analysis was added in the revised manuscript as suggested (lines 188-191).

Table 1 lists 13 genes that share sequence homology with genes known to be involved in biofilm formation. However, there is a notable absence of discussion or explanation regarding the roles these genes may play in the biofilm-forming regulatory network of D. radiodurans. The study must investigate whether any of these genes influence biofilm development. For instance, the gene dr_0408 was observed to be downregulated in the drRRA mutant compared to the wild-type strain (lines 325–326). The role and impact of DR_0408 in biofilm formation should be examined and elaborated upon in the manuscript.

A: To grow in various harsh environments, extremophiles have developed extraordinary strategies, such as biofilm formation, an extremely complex and progressive process, but the biofilm formation-related genes and underlying mechanism of D. radiodurans R1 are poorly elucidated. To explore potential genes involved in biofilm formation of D. radiodurans R1, we identified 409 differentially expressed genes during D. radiodurans biofilm development by using RNA-seq. Among these differentially expressed genes, only 13 matched genes share sequence homology with genes related to biofilm formation of E. coli, P. aeruginosa and S. aureus (Table 1), but these matched genes are not the key regulators of biofilm formation based on the previous study. In view of the above, we added the references in Table 1 and the description of these genes in the revised manuscript (lines 193-195).

It is widely known that TCSs play a critical role in bacterial adaptation and survival under various extreme environmental conditions. Therefore, in this study, we focused on the response regulator DrRRA and identified that DrRRA, as a facilitator of biofilm formation, could directly stimulate the transcription of the biofilm-related gene drBON1. For the 13 homology genes in Table 1, both DR_0408 and DR_0888 are differentially expressed in the drRRA mutant under biofilm development condition at all stages (Table S2), but the putative DrRRA binding site was only discovered in the promoter of dr_0408, indicating that expression of dr_0408 might be directly regulated by DrRRA. To address this comment, we supplemented the description of the two genes (lines 251-255) as follows: Furthermore, only 2 of the 13 homology genes presented in Table 1, dr_0408 and dr_0888, are differentially expressed in the drRRA mutant during the four stages of biofilm development. However, the DrRRA-binding site was only discovered in the promoter of dr_0408, indicating that DrRRA plays a direct regulatory role in the expression of dr_0408.

Section 2.4 elucidates the role of DrRRA in D. radiodurans' biofilm formation. I find the rationale for focusing on this gene to be insufficiently substantiated. While the manuscript acknowledges the importance of two-component systems (TCSs) in bacterial adaptation and survival under harsh conditions and identifies the response regulator DrRRA as a key factor in D. radiodurans' radioresistance (lines 205-207), this alone does not adequately justify its selection for study. It would seem more logical to examine DR_0997 (DdrI), as listed in Table 1. This gene is a homolog of one involved in biofilm formation and is a transcription regulator linked to D. radiodurans' stress response (DOI:10.1371/journal.pone.0155010). Why didn't the authors investigate DdrI?

A: Two-component systems connect the environmental signals and cellular response, which is significant for biofilm development. It was reported that the DR_0997 (DdrI) expression is regulated by DrRRA (Wang l et al., 2008, DrRRA: a novel response regulator essential for the extreme radioresistance of Deinococcus radiodurans. Mol Microbiol, 67(6):1211-22). Furthermore, among the 409 DEGs during biofilm development, 228 genes are affected by drRRA mutation (Wang L et al., 2008, A novel response regulator essential for the extreme radioresistance of Deinococcus radiodurans. Mol Microbiol, 67(6):1211-22). These data indicated that DrRRA might be an essential regulator of its biofilm formation under extreme environmental conditions. Therefore, here, we focused on the response regulator DrRRA. In addition, we also attempted to construct the dr_0997 mutant, but was unsuccessful. The functions of DR_0997 in biofilm formation still remain to be elucidated in future.

A comparative analysis of the 409 DEGs during biofilm formation with those affected by drRRA mutation (DOI: 10.1111/j.1365-2958.2008.06113.x) would be valuable. If there is an overlap between these genes, it could provide a more compelling reason for selecting DrRRA. Additionally, the target gene drBON1 choice also appears to have a weak rationale. The paper notes that the expression level of dr_0392, a gene encoding a membrane-binding protein, was significantly downregulated in the drRRA mutant (lines 245-247). However, the expression levels of dr_1314 and dr_1199 were reduced even more (Table S2). Notably, dr_1314 is part of a heat shock regulon (DOI: 10.1128/JB.187.10.3339-3351.2005). Hence, it would be prudent for the authors to investigate the effects of other DrRRA-regulated genes on biofilm development to provide a more robust and comprehensive understanding of the underlying mechanisms.

A: Thank you for pointing this issue out. Among these 409 DEGs, 228 genes are affected by drRRA mutation (Wang l et al., 2008, DrRRA: a novel response regulator essential for the extreme radioresistance of Deinococcus radiodurans. Mol Microbiol, 67(6):1211-22), and we have listed these overlapping genes in the new Table S5.

We agree with the reviewer that the expression levels of dr_0392, dr_1314 and dr_1199 were affected by DrRRA during the four stages of biofilm development (Table S2), and the putative DrRRA-binding motif was also discovered in the promoters of the three genes. The previous study showed that DR_0392 is involved in the resistance of D. radiodurans R1 to NaCl stress and γ-irradiation (Im S et al., 2013, Transcriptome analysis of salt-stressed Deinococcus radiodurans and characterization of salt-sensitive mutants. Res Microbiol, 164(9):923-32). In this study, we found that extremely high concentrations of NaCl or sorbitol could induce biofilm formation. Thus, we could infer that DR_0392 might be involved in the biofilm formation under extreme environmental conditions and exclusively focus on DR_0392.

The manuscript currently presents three distinct topics: (i) the induction of biofilm formation by osmotic stress, (ii) transcriptome analysis during biofilm development under normal conditions, and (iii) the effect of DrRRA on biofilm formation, but fails to integrate them cohesively. I recommend thoroughly revising the Discussion section to create a clear narrative thread that connects these topics, highlighting how each contributes to our understanding of D. radiodurans biofilm formation. Furthermore, the manuscript should be revised to consider the previously reported composition of D. radiodurans' EPS (doi.org/10.1038/s41598-019-56141-3). Specifically, lines 327–332 should be updated to reflect the current study's results and interpretations.

A: We agree and have modified the discussion about the three distinct topics presented in this manuscript as suggested. On the one hand, discussed the biofilm under osmotic stress condition. On the other hand, making the comprehensive discussion about the transcriptional data of both WT and the drRRA mutant (lines 314- 330).

The transcription of DR_A0033, the previously reported composition of D. radiodurans' EPS (doi.org/10.1038/s41598-019-56141-3), was no significant difference during biofilm development in WT, but its expression was affected in drRRA mutant (log2(fold change) value =1.99, (p< 0.05) in T66/M66). Thus, the underlying mechanism of DR_A0033 regulated by DrRRA during biofilm formation could be explored in future. To address this comment, we modified the discussion (lines 369-375) as follows: Furthermore, DR_A0033, a ExoP-like protein, was involved in the exopolysaccharides synthesis of D. radiodurans, and compared to the WT, biofilm biomass of the dr_A0033 mutant is reduced [63]. However, its transcription is not differential expressed during biofilm development in WT, but compared to the WT, its expression was affected in drRRA mutant (log2(fold change) value =1.99, (p< 0.05) in T66/M66). Thus, the underlying mechanism of dr_A0033 and DrRRA during biofilm formation could explore in future.

Minor comments

- line 66: Deinococcus metalilites à Deinococcus metallilatus

A: Done (line 66).

- line 76: replace '31' and '32' with more relevant references

A: Done (line 76).

- line 88: abiotic stress resistance à abiotic stress

A: Done (line 91).

- line 208: thatno à that no

A: Done (line 221).

- Fig. 8C: what is the dr_0891?

A: we have added the detailed information on Δdr_0891 in Table S3. Furthermore, the description of dr_0891 was added in the revised manuscript ((lines 286-287) as follows: dr_0891 encodes a response regulator in D. radiodurans, but the biofilm formation abil-ity of the dr_0891 mutant is similar to WT (data not shown).

- line 266: delete the reference '30'

A: Done (line 286).

- line 299: card à card

A: Done.

- line 329: dr_a0047 à dr_A0047

A: Done (line 366).

- line 375: correct ‘220 r/min’ and ‘OD600’

A: We have modified this mistakes (line 418).

- line 430: specifiy the kinds of UV rays (UVA, B or C?) and the device used.

A: Done (lines 488-490).

Reviewer 2 Report

Comments and Suggestions for Authors

General considerations

In the manuscript entitled “Development and regulation of extreme biofilm formation of Deinococcus radiodurans R1 under extreme environmental conditions”, the authors suggest that biofilms are a good survival strategy for D. radiodurans under extreme environmental conditions. This hypothesis was supported by the faster formation of mature biofilms when D. radiodurans was subjected to high concentrations of NaCl and sorbitol, and because the biofilms increased the cell resistance against UV radiation, extreme oxidative stress and sorbitol stress. In addition, the authors suggest that the response regulator DrRRA is an essential component that facilitates the biofilm formation, and directly regulates the drBON1 gene, that was also associated with biofilm formation in D. radiodurans. In general, the manuscript is well structured, objective, clear and concise. However, there are some results that are not well explored, and some decisions that do not seem obvious.

Minor comments

Legend of Figures 1, 2, 3, 6, 8 - In the legend of these figures, please, mentioned what is represented by the error bars (the standard deviation?). 

Line 118 – Legend of Figure 2. Please clarify between which samples the statistical significance represented above each column was determined.

Table 2 and S2 - Please define in the table legend/notes what the meaning of “ND” is, and“N/A” in table S2.

Line 208: “showed thatno significant” please replace by “showed that no significant”

Line 216: “formation of in the extremophilic” please replace by “formation of the extremophilic”.

Line 219: Why do you use the term microRNA to refer to the messenger RNA drRRA?

Figure 8b – Please define what is the meaning of “CK” in the legend of this figure.

Line 299: “and card mutants” please replace by “and carD mutants”

Table S3 – Please add the reference of the Deinococcus radiodurans wild type strain in the table, and some information about their origin. Regarding the drRRA mutant, please add the Reference 30 in the table.

Major Comments

1.      Regarding the description of the RNA-seq procedure, the methodology used to analyze the data is missing, as well as the statistical analysis performed. Where are the RNA-seq raw data submitted and available?

2.      Although several RNA-seq experiments were performed, the authors chose to analyze genes already identified as stress response related genes? Why? Did you try to explore and study other genes?

3.      Regarding the drBON1 gene, it is described in table S2 as a hypothetical protein DR_0392. Several other genes could be selected regarding the results of these table. Why did you focus only in the drBON1 gene?

4.      Regarding the EMSA assay, could you please provide the original image of the EMSA? In this experiment, very high concentrations of DrRRA protein were used, which may compromise the biological significance of the inferences made. Which are the DrRRA protein levels in Deinococcus radiodurans? Could you confirm the direct interaction of DrRRA with the drBON1 promoter by other method?

Round 2

Reviewer 1 Report

Comments and Suggestions for Authors

Although the authors have tried to address the concerns I previously raised, some of their responses are insufficient to fully alleviate these concerns.

1. IrrE (PprI) and PprM are well-established as key players in D. radiodurans' resistance to various stresses. However, this manuscript introduces CarD and DtrA/B for the first time. Consequently, it is premature to assert that CarD and DrtA/B are pivotal in controlling D. radiodurans' stress resistance based solely on 'unpublished data.' To substantiate such claims, this manuscript should include data demonstrating the phenotypes of these mutants under stress conditions, along with the relevant methodologies (such as mutant construction).

2. Additionally, concrete evidence is required to elucidate the specific regulatory pathway of CarD in biofilm formation. In my initial review, I emphasized the importance of identifying the primary trigger for biofilm formation, be it osmotic stress or growth arrest. It's crucial to compare the growth profiles of carD mutants with those of the wild-type under osmotic conditions. The manuscript mentions ppGpp as a potential key to understanding CarD-dependent biofilm formation. Does the carD mutation impact the expression of genes dr_1838 (relA) and dr_1631 (relQ), which are involved in the synthesis and hydrolysis of (p)ppGpp, particularly under conditions of high osmotic stress? Given that growth arrest typically leads to elevated levels of (p)ppGpp, this implies that CarD's involvement in biofilm formation could be associated with a pathway related to growth regulation rather than being specific to the osmotic stress response.

Reviewer 2 Report

Comments and Suggestions for Authors

Dear authors,

Thank you for revising the manuscript and for your responses.

Minor Comments:

Line 116: "experimentsare" please replace by "experiments are".

Line 472: D. radiodurans must be italicized.

Round 3

Reviewer 1 Report

Comments and Suggestions for Authors

Kindly incorporate the results (figures R1 to R4) into Section 2.1 (lines 125-132) as supplementary materials and remove lines 78-80 from the Introduction. Furthermore, please specify the DR locus tags for these genes.
